# Effects of Liquid Parameters on Liquid-Filled Compartment Structure Defense Against Metal Jet

**DOI:** 10.3390/ma12111809

**Published:** 2019-06-04

**Authors:** Xudong Zu, Wei Dai, Zhengxiang Huang, Xiaochun Yin

**Affiliations:** 1School of Mechanical Engineering, Nanjing University of Science and Technology, Nanjing 210094, China; huangyu@njust.edu.cn; 2Beijing Special Vehicle Research Institute, Beijing 100072, China; 13601366894@139.com; 3Department of Mechanics and Engineering Science, Nanjing University of Science and Technology, Nanjing 210094, China; yinxiaochun@njust.edu.cn

**Keywords:** liquid parameter, depth of penetration, shaped charge jet, liquid-filled structure

## Abstract

The effect of liquid parameters on the defense capability of the liquid-filled compartment structure (LFCS) of a shaped charge jet (SCJ) is quantified using dimension analysis of experiments on the reduced depth of SCJ penetration, which is disturbed via the LFCS with different liquids. The effects of three parameters, namely, liquid density, sound velocity and dynamic viscosity, on LFCS defense for SCJ are discussed quantitatively. Dynamic viscosity exerts the most important effect on LFCS disturbance of SCJ penetration, followed by liquid density. Meanwhile, sound velocity causes a negligible effect on LFCS disturbance of SCJ when the hole diameter in LFCSs are short. LFCSs offer excellent protection as they can significantly reduce the penetration capability of SCJ. Thus, LFCSs can be used as a new kind of armor for defense against SCJ.

## 1. Introduction

Liquid-filled compartment structure (LFCS) is an excellent structure that can be used on the side tanks of ships and as an additional liquid-filled cover armor of armored vehicles to resist incoming penetrative projectiles. As a kind of armor, LFCS can disturb the stability of high-speed projectiles during penetration. However, no reliable model can be used to accurately describe the interaction between a LFCS and projectiles and evaluate the effects of the former on the latter. High explosive anti-tank (HEAT) cartridges represent an important kind of ammunition that is widely used to attack armored target. A LFCS can reduce the penetration capability of shaped charge jet (SCJ) from HEAT cartridges. The properties of liquid materials (e.g., density, sound velocity and dynamic viscosity) play an important role in the way LFCSs disturb the stability of the SCJ and reduce their penetration capability [1]. Therefore, this study establishes a mode that includes the parameters of liquid materials in calculating the effects of the LFCS to the penetration capability of SCJ through dimensional analysis.

In general, during projectile impact (fragments or bullets) in a liquid container, a cavity is formed by projectile drag, and a bubble is created by a hydrodynamic ram event induced by projectile penetration at ballistic speed in a confined geometry filled with liquid; this condition reduces the kinetic energy of the projectile and changes the motion trail. Most researchers pay considerable attention to expansion and collapse cavity and the deformation and destruction of containers. Lecysyn [2] analyzed shock wave propagation, cavity formation and energy loss of fluid-filled tanks under high-speed projectile impact through experimental work and theoretical modeling. The experimental phenomena can be described by a theoretical model. Disimile [3] undertook a detailed study on hydrodynamic ram and the destructive interference offered by a pressure mitigation system through firing a spherical projectile onto a water tank. The results showed that the hydrodynamic ram effect can be reduced with an appropriate assembly design. Charles [4] used numerical methods to model the expansion and collapse of cavities that develop in containers after impact and arrest by water with ballistic projectiles. A numerical model was used to evaluate the proportion of kinetic energy of the bullet that was converted into internal, fluid kinetic and latent energy of water, which can finally be transferred to a deformable fuel tank structure. Fourest et al. [5,6] analyzed bubble dynamics through a hydrodynamic ram in a pool and liquid-filled container by using the Rayleigh–Plesset equation. The results showed that liquid compressibility influences the dynamics of confined bubbles, and that the gas pressure in bubbles exerts minimal influence on bubble dynamics and the hydrodynamic loads applied to the structure. Varas [7] studied plastic deformation and the pressure of water-filled aluminum square tubes filled with fluid at different levels and subjected to the impact of a spherical steel projectile with various velocities. They observed that the momentum normal to the wall is the most important factor influencing tube deformation. Varas [8] investigated liquid pressure, wall displacement of a square tube and cavity evolution at different impact velocities created by a steel spherical projectile impacting a partially water-filled aluminum square tube through simulations and experiments. The results proved that the arbitrary Lagrange–Euler technique can reproduce the stages of a hydrodynamic ram in partially filled tubes and cavity evolution, which is the main cause of final tank deformations. Artero-Guerrero [9] analyzed pressure at different points and strains on the tube wall and cavity evolution under a projectile with different velocities to impact a liquid-filled woven carbon fiber-reinforced plastic tube through simulation and experimental methods. The numerical results showed that the relationship between the magnitude of pressure pulse and impact velocity quadratically increased, and that the maximum cavity size linearly increased. Deletombe [10] observed and measured the dynamic evolution of the geometry of the wake behind and the cavity around a projectile with a 7.62 mm bullet shot liquid-filled container by using high-speed digital image cameras. Kong [11] investigated the penetration of single and double fragments into a liquid-filled cabin. The simulation results showed that the peaks of shock wave pressure from the impact of double fragments are twice higher than those in single fragments.

Several researchers also considered the effects on projectiles. Sauer [12] investigated the impact response of projectiles on fluid-filled containers through numerical modeling, in which container rupture, water spread, and residual velocity were studied through 2D axisymmetric hydrocode simulations. The results showed that the adaptive smoothed particle hydrodynamics approach can be accurately used to calculate deformation, water spread and residual velocity. Uhlig [13] examined the interplay of an eroded target material and the remaining projectile, wherein copper rods perforated the liquid-filled channels that were circumferentially confined by steel cylinders. Channel width was found to play a more important role during penetration in low-density materials than in high-density materials.

Shaped charge warheads are the main warheads used to attack ships and armored vehicles. Researchers analyzed the effects of liquid-filled structures on SCJ. Held [14] modified Szendrei equations and obtained the reaming equations of jet penetration in water through high spatial and temporal resolutions and profile streak technology. Lee [15] explored the penetration of jet particles by using water in high-speed photography and X-ray experiments. The researchers observed that the contour of bubbles formed no smooth curve, but that wave packets and jet particles only arrived at the penetration bottom when the foregoing particles were consumed completely during penetration. Huang et al. [16,17] established a mechanical model of a diesel oil-filled hermetic structure that interferes with a jet and obtained the expressions of the interfered velocity range. They also verified the theoretical model via X-ray experiments.

In the present study, liquid parameters, such as density, sound velocity, and dynamic viscosity, are considered. The relationship between the defense capability of LFCSs, directly shown as the reduced depth of penetration (DOP) by the LFCS and SCJ and target, is obtained through dimensional analysis. Coefficient values are fitted on the basis of the experimental results. The effects of each liquid parameter on reduced DOP is discussed.

## 2. Physics Phenomena and Principles

Take for example a HEAT attack on a tank with LFCS as additional armor. The effect of the LFCS on the SCJ of the HEAT cartridge is shown in Figure 1. The LFCS as an additional armor is always fixed on the main armor by bolts which establishes a certain distance between the LFCS and the main armor (Figure 1II). When the HEAT cartridge attacks the tank with LFCS as additional armor (Figure 1I), the fuse detonates the explosive in the HEAT cartridge, and SCJ is formed and penetrates the LFCS after the node of the HEAT cartridge impacts the LFCS. In the process of the SCJ penetrating, the disturbed jet drifts off from the penetrating axis (Figure 1III).

In the present research, the LFCS consists of a main target body and a front plate. The main target body comprises bulk #45 steel drilled with matrix cylindrical blind holes. The materials of the main target body and front plate are #45 steel and Q235 steel, respectively. Figure 2 shows the target size.

The schematic of the interaction between the SCJ and LFCS is shown in Figure 3. The basic principle [16,17] between the LFCS and the SCJ is presented as follows:

When the SCJ penetrates the liquid in the LFCS, a shock wave is assumed to be formed from the tip of the SCJ, because the penetration velocity is faster than the sound velocity of the liquid (t_1_). The shock wave is a conically diffused spherical wave, and its propagation direction is perpendicular to the wave front along its normal direction (from A to B along line І). The initial shock wave quickly reaches the sidewall of the structure that reflects the shock wave (from B to C along line ІІ). Reflected wave propagation prevents the reaming process of the subsequent jet. If the surface stress of the reflected shock wave is greater than the expansion stress, then the liquid enters a convergence process. The convergence liquid under the reflected wave disturbs the SCJ (t_2_) and makes that part of the SCJ drift off from the penetrating axis. 

In previous research, only one kind of liquid diesel used in the LFCS and the sound velocity of the liquid were considered. Moreover, the density and dynamic viscosity of the liquid material were ignored. However, density and dynamic viscosity play important roles in the LFCSs disturbance of the stability of the SCJ. In fact, effects are complicated when the LFCS interacts with the SCJ (high temperature of 800–1000 °C, high pressure of 10^9^–10^10^ Pa). However, the whole interaction is difficult to describe. Therefore, dimension analysis is suitable for the study of the effects of liquid parameters on LFCS defense against SCJ.

## 3. Dimension Analysis

In broadening the range of applications, dimensional analysis requires no general governing equations. To simplify the model and show the major influencing factors, this study makes the following assumptions:(1)The density of the SCJ shows no change;(2)The SCJ is continuous;(3)Liquid vaporization is ignored as the whole penetration process is extremely short.

Hence, when the SCJ penetrates the liquid-filled structure, the independent factors that affect the penetration capability of the SCJ are as follows:(1)Parameters of SCJ: density $\rho_{j}$, velocity of jet tip $v_{j0}$, velocity of jet tail $v_{jt}$, initial length of the jet $l_{0}$ and reduced DOP owing to the disturbed SCJ $P_{dis}$;(2)Liquid parameters: density $\rho_{l}$, sound velocity $C_{l}$ and dynamic viscosity μ;(3)Parameters of compartment structure: inner diameter D, height of compartment H, sound velocity of compartment material $C_{t}$ and density of compartment material $\rho_{t}$.

Table 1 provides the symbols, units and dimensions of the parameters used in this study.

The penetration capability of the disturbed SCJ (reduced DOP) can be determined as follows:(1)$$P_{dis}=f\left(\rho_{l},C_{l},\mu ,v_{j0},v_{jt},l_{0},\rho_{j},C_{j},D,H,C_{t},\rho_{t}\right)$$
where $\rho_{l}$, $C_{l}$ and $l_{0}$ are independent dimension parameters. On the basis of the π theorem, k = 3, Equation (1) becomes dimensionless.
(2)$$\frac{P_{dis}}{l_{0}}=f^{\prime }\left(\frac{\mu }{\rho_{l}\cdot C_{l}\cdot l_{0}},\frac{v_{j0}}{C_{l}},\frac{v_{jt}}{C_{l}},\frac{\rho_{j}}{\rho_{l}},\frac{C_{j}}{C_{l}},\frac{D}{l_{0}},\frac{H}{l_{0}},\frac{C_{t}}{C_{l}},\frac{\rho_{t}}{\rho_{l}}\right)$$

Let $\pi_{1}=\frac{\mu }{\rho_{l}\cdot C_{l}\cdot l_{0}}$, $\pi_{2}=\frac{v_{j0}}{C_{l}}$, $\pi_{3}=\frac{v_{jt}}{C_{l}}$, $\pi_{4}=\frac{\rho_{j}}{\rho_{l}}$, $\pi_{5}=\frac{C_{j}}{C_{l}},\;\pi_{6}=\frac{D}{l_{0}}$, $\pi_{7}=\frac{H}{l_{0}}$, $\pi_{8}=\frac{C_{t}}{C_{l}}$, $\pi_{9}=\frac{\rho_{t}}{\rho_{l}}$.

The variables are reversed as follows to simplify their relation and clarify their physical significance:

$\pi_{1}^{\prime }=\pi_{1}/\pi_{6}$, $\pi_{3}^{\prime }=\pi_{3}/\pi_{2}$, $\pi_{5}^{\prime }=\pi_{2}/\pi_{5}$, $\pi_{8}^{\prime }=\pi_{8}/\pi_{5}$, $\pi_{9}^{\prime }=\pi_{9}/\pi_{4}$. Then, Equation (2) is transformed into the following:(3)$$\frac{P_{dis}}{l_{0}}=g^{\prime }\left(\frac{\mu }{\rho_{l}\cdot C_{l}\cdot D},\frac{v_{j0}}{C_{l}},\frac{v_{jt}}{v_{j0}},\frac{\rho_{j}}{\rho_{l}},\frac{v_{j0}}{C_{j}},\frac{D}{l_{0}},\frac{H}{l_{0}},\frac{C_{t}}{C_{j}},\frac{\rho_{t}}{\rho_{j}}\right)$$

Given that the shaped charge and target are confirmed, and that $\frac{v_{jt}}{v_{j0}}$, $\frac{v_{j0}}{C_{j}}$, $\frac{D}{l_{0}}$, $\frac{H}{l_{0}}$, $\frac{C_{t}}{C_{j}}$, $\frac{\rho_{t}}{\rho_{j}}$ are constant, Equation (3) can be simplified as follows:(4)$$\frac{P_{dis}}{l_{0}}=\Phi \left(\frac{\mu }{\rho_{l}\cdot C_{l}\cdot D},\frac{v_{j0}}{C_{l}},\frac{\rho_{j}}{\rho_{l}}\right)$$

In Equation (4), $\frac{\mu }{\rho_{l}\cdot C_{l}}$ is the ratio of dynamic viscosity to the inertia force of the moving liquid and it characterizes the flow characteristic of the liquid. $\frac{v_{j0}}{C_{l}}$ is the ratio of the velocity of jet tip to the sound velocity of the liquid and it characterizes the shock wave strength when the jet penetrates the liquid. $\frac{\rho_{j}}{\rho_{l}}$ is density ratio of the jet to the liquid. Equation (4) can be written as a power function.
(5)$$\frac{P_{dis}}{l_{0}}=A\cdot {\left(\frac{\mu }{\rho_{l}\cdot C_{l}\cdot D}\right)}^{\alpha }\cdot {\left(\frac{v_{j0}}{C_{l}}\right)}^{\beta }\cdot {\left(\frac{\rho_{j}}{\rho_{l}}\right)}^{\gamma }$$

Referring to the 1D hydrodynamic penetration model, the DOP is equal to the length of the jet multiplied by the square root of the density ratio of the jet to the target. Thus, the number of γ may be 1/2. So $P_{dis}$ can be written as follows:(6)$$P_{dis}=A\cdot {\left(\frac{\mu }{\rho_{l}\cdot C_{l}\cdot D}\right)}^{\alpha }\cdot {\left(\frac{v_{j0}}{C_{l}}\right)}^{\beta }\cdot l_{0}\sqrt{\frac{\rho_{j}}{\rho_{l}}}$$
where A, α and β represent the coefficients to be confirmed by experimental results. $l_{0}\sqrt{\frac{\rho_{j}}{\rho_{l}}}$ refers to the 1D hydrodynamic penetration jet model.

## 4. Experimental Research

### 4.1. Shaped Charge and Liquid

For general research, the standard Φ56 mm shaped charge is adopted (Figure 4). The jet tip velocity is 6453 m/s, and that of the tail is 1179 m/s. The length of the jet at 80 mm standoff is equal to 111.5 mm [18].

In the experiments, five kinds of liquid are selected: water, #0 diesel, polyethylene glycol with molecular weights of 200 and 400 (PEG 200 and PEG 400, respectively) and polypropylene glycol with molecular weight of 6000 (PPG 6000). The test method and conditions are the same as those of Newtonian fluid in reference [19]. Table 2 displays the main material parameters.

### 4.2. Experimental Setup and Results

The compartments of the LFCS were filled with liquid and sealed by the front plate through bolts before experiments. The liquid in the LFCS could not flow out. The DOP experiments on the standard shaped charge were performed initially to ensure the reduced penetration capability (reduced DOP) of the SCJ. Then, the residual DOP experiments on SCJ penetration involving the LFCS with different liquid materials were carried out. In all the experiments, the distance from the bottom of the shaped charge to the surface of the target was the same.

The setup of the DOP experiment on the shaped charge at a standoff of 254 mm is shown in Figure 5. The setup of the residual DOP experiment is shown in Figure 6. The standard shaped charge connects to the target liquid-filled structure through an 80 mm standoff cylinder, whereas the #45 steel target connects to a 90 mm interval cylinder. The DOP experiment in which the standoff is 254 mm (80 mm + 84 mm + 90 mm) is carried out to obtain the penetration capability of the SCJ with the same distance as that when the front of the steel target lacks a liquid composite armor. Every DOP experiment is repeated twice. 

The results of the DOP experiment were 254 and 258 mm. Therefore, the average result is 256 mm. One of the witness targets was cut into two pieces via wire electrode cutting, as shown in Figure 7.

Table 2 illustrates the residual DOP (P_res_) results. Figure 8, Figure 9, Figure 10, Figure 11 and Figure 12 display the DOP in the witness target with different liquids.

Table 3 and Figure 8, Figure 9, Figure 10, Figure 11 and Figure 12 show that the LFCS can considerably decrease the DOP of the SCJ. However, comparison results in Table 2 and Table 3 reveal that the LFCS can reduce the penetration capability of the SCJ obviously. Comparing the single parameter of the liquid with P_dis_, shows that P_dis_ increases with the dynamic viscosity of the liquid, except for PPG6000. Moreover, P_dis_ increases with the density of the liquid, except for diesel. No relationship exists between P_dis_ and the sound velocity of the liquid. Thus, the parameters of the liquid in the LFCS show no notable relation with P_dis_.

Comparing Figure 8, Figure 9, Figure 10, Figure 11 and Figure 12 with Figure 7 reveals that P_res_ sharply drops by 68% after the SCJ penetrates the LFCS. The penetration holes on the witness target are not as straight as that when SCJ penetrates the target directly. In addition, the material of the jet was observed on the wall of the penetration hole of the witness target. These results are mainly explained as follows. The stability of the SCJ is disturbed when it pierces the LFCS. Parts of the SCJ segments veer off the axis before reaching the bottom of the penetration hole penetrated by segments. These parts of the SCJ segments cause the reduction of P_res_. Although parts of the SCJ segments drift off the axis, they can still reach the bottom of the penetration hole penetrated by previous segments. These parts of the SCJ segments have penetration capability, but they cause the penetration holes to bend. 

According to the liquid parameters in Table 2 and the reduced DOP in Table 3, the experimental result involves five groups, and Equation (6) only comprises three coefficients. Thus, the coefficient of Equation (6) can be obtained by Origin 8.5. Equation (6) can be computed as follows:(7)$$P_{dis}=0.62192\cdot {\left(\frac{\mu }{\rho_{l}\cdot C_{l}\cdot D}\right)}^{5.77e-3}\cdot {\left(\frac{v_{j0}}{C_{l}}\right)}^{-3.07e-3}\cdot l_{0}\sqrt{\frac{\rho_{j}}{\rho_{l}}}.$$

## 5. Remarks

In the analysis of the effects of liquid parameters, one parameter is changed, and another two are fixed. By considering a parameter change, the change in the disturbed DOP P_dis_ is obtained (Figure 13, Figure 14 and Figure 15).

Figure 13 shows that with an increase in liquid density, the P_dis_ also decreases. For the five kinds of liquid, when the liquid density increases from 800 kg/m^3^ to 1500 kg/m^3^, and when the reduced ratio of P_dis_ gradually decreases, P_dis_ drops by 27.2%. This result means that at a low liquid density in the compartment structure, the structure can disturb a wide jet velocity range, thus the P_dis_ increases. 

The density characterizes the inertia of the liquid. The inertia increases with a rise in density. The state of motion does not easily change with an increase in density under the same stress. Thus, with an increase in density, the difficulty of liquid reflow in the LFCS rises; hence, the effect of the LFCS on the SCJ decreases, which indicates that liquid density increases with a decrease in P_dis_.

Figure 14 shows that with increased sound velocity of the liquid, the P_dis_ slightly changes. For the five kinds of liquid, when the sound velocity of the liquid increases from 1000 m/s to 3000 m/s, P_dis_ reduction does not exceed 0.5%, indicating that the effect of sound velocity of the liquid on reduced DOP can be ignored.

When the sound velocity of the liquid increases, the time between shock wave formation and the reflection of the shock wave on the SCJ is reduced. In this way, the LFCS can disturb the high velocity range jet. However, in the current research, the diameter of the hole in the LFCS is considerably short, hence, the effect of the sound velocity of the liquid is minimal.

Figure 15 shows that the P_dis_ exhibits a linearly increasing relation with the logarithm of dynamic viscosity of the liquid. Therefore, the high liquid viscosity in the compartment structure can substantially affect the SCJ and effectively reduce the penetration capability of the jet.

The dynamic viscosity of the liquid characterizes the viscosity of such liquid. Under zero sliding condition, an increase in the dynamic viscosity of the liquid boosts the effect of liquid reflow in the LFCS to the SCJ; that is P_dis_ increases with a rise in the dynamic viscosity of the liquid.

## 6. Conclusions

In this study, the effects of liquid parameters and P_dis_ are studied. Conclusions can be formulated as follows:

(1) The reduced DOP of the SCJ can be described by Equation (6) when the liquid in the LFCS is Newtonian liquid. In addition, when the parameters of the SCJ and LFCS are fixed, the coefficient is easily confirmed by experimental results.

(2) Based on Equation (7), the calculated results point out that the dynamic viscosity of liquid exerts the most important effect on LFCS disturbance of SCJ, with liquid density having the second most important impact. Moreover, sound velocity causes a negligible effect on LFCS disturbance of SCJ when the holes’ diameter in the LFCS are short. 

In obtaining an LFCS with high defence capability, that is, the LFCS can further reduce the DOP of the SCJ, the liquid in the structure should feature high dynamic viscosity and low density.

(3) LFCSs can disturb the stability of SCJ and reduce their DOP. These structures can thus be used as a new kind of armor.

## Figures and Tables

**Figure 1 materials-12-01809-f001:**
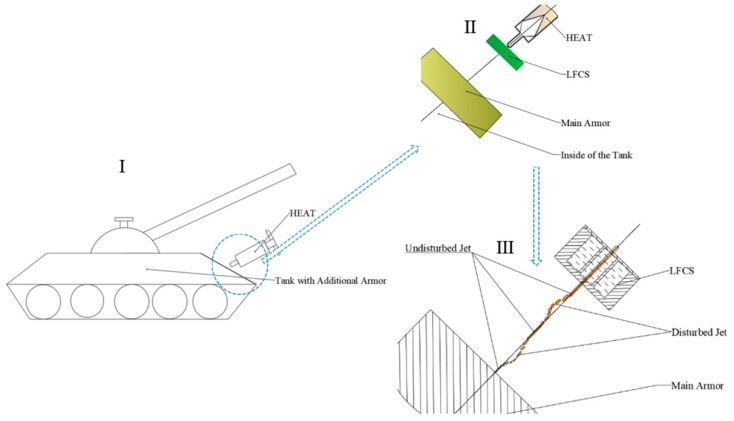
Effects of the liquid-filled compartment structure (LFCS) on the shaped charge jet (SCJ) of high explosive anti-tank (HEAT) cartridge.

**Figure 2 materials-12-01809-f002:**
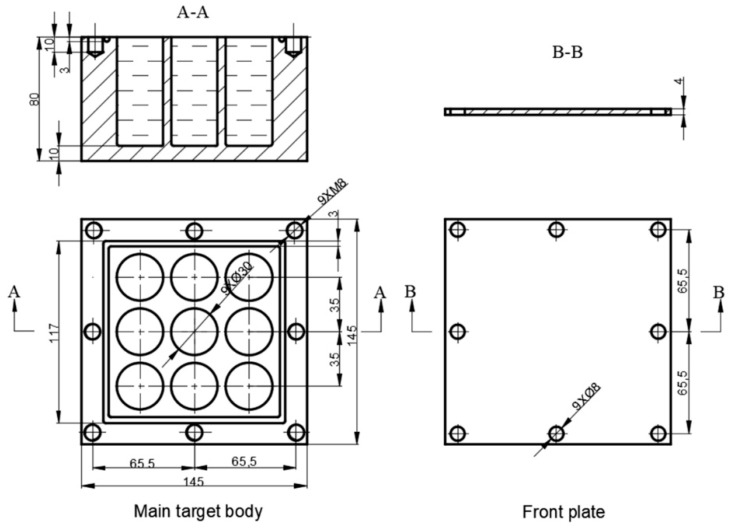
LFCS (all dimensions in mm).

**Figure 3 materials-12-01809-f003:**
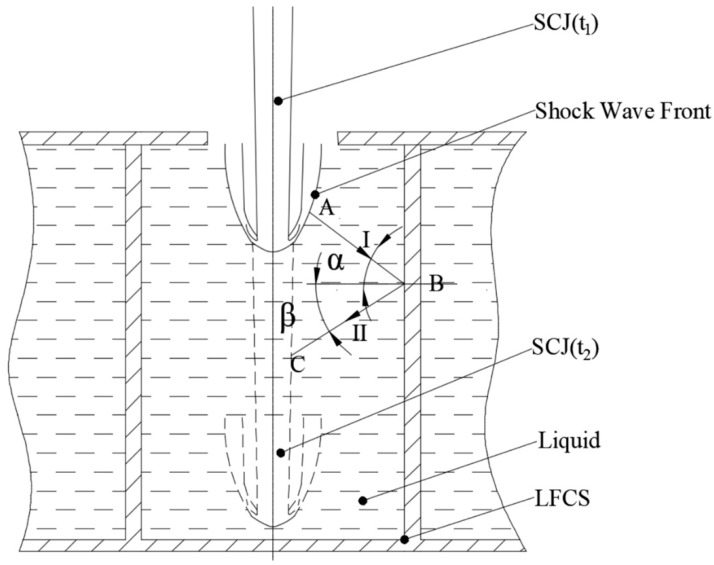
Interaction between the SCJ with the LFCS.

**Figure 4 materials-12-01809-f004:**
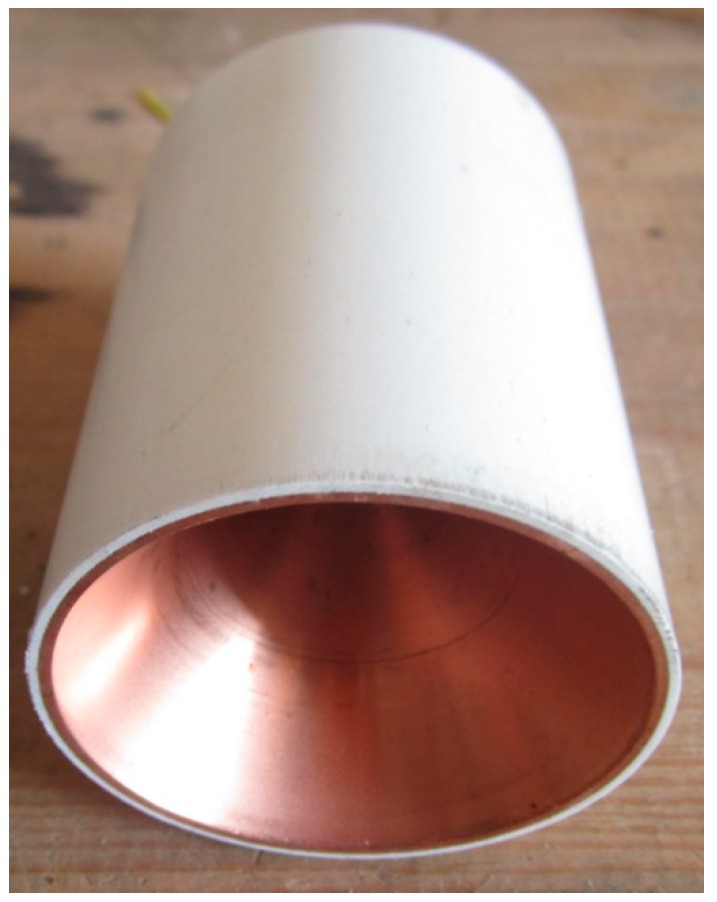
Standard shaped charge.

**Figure 5 materials-12-01809-f005:**
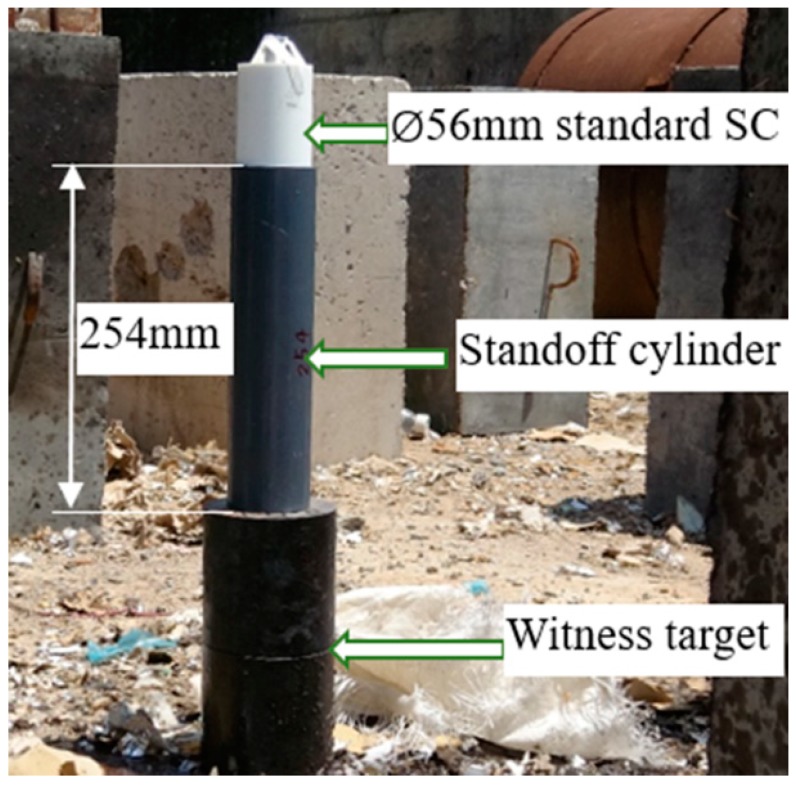
Depth of penetration (DOP) experimental setup.

**Figure 6 materials-12-01809-f006:**
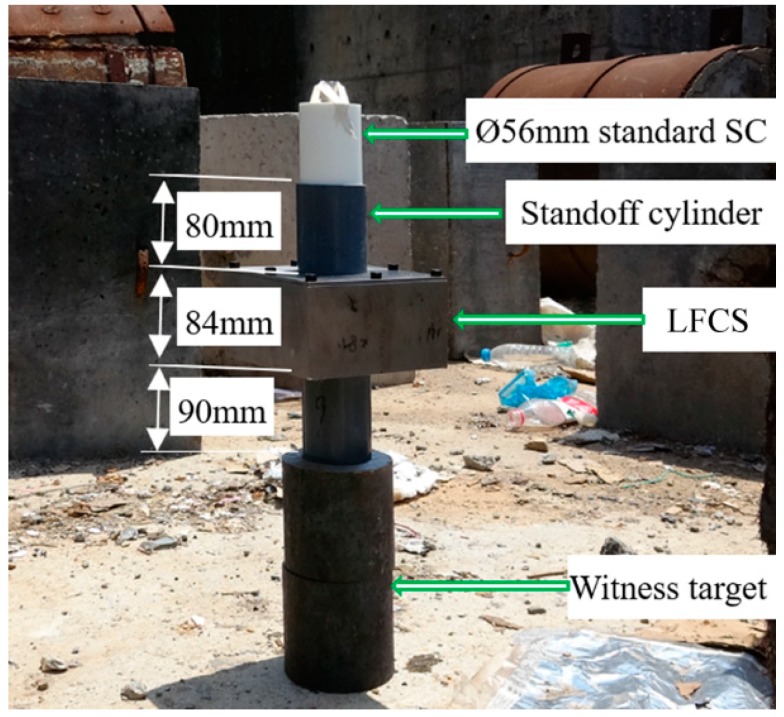
Experimental setup.

**Figure 7 materials-12-01809-f007:**
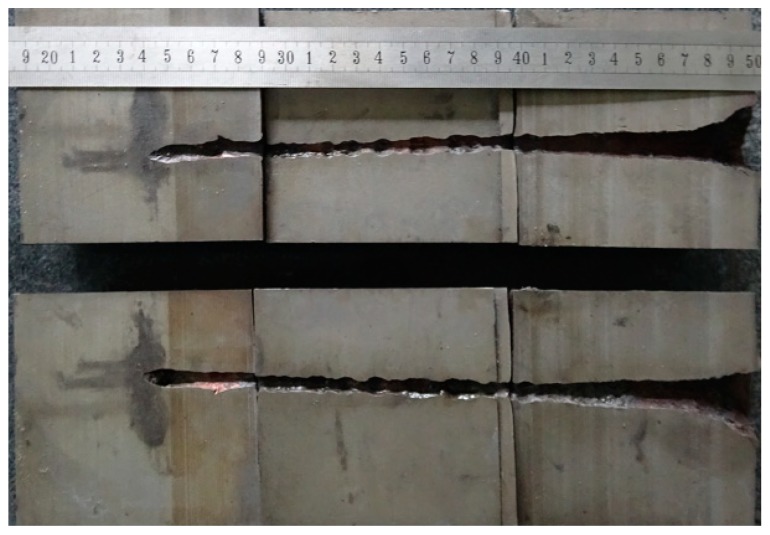
Split image of witness target obtained by linear cutting.

**Figure 8 materials-12-01809-f008:**
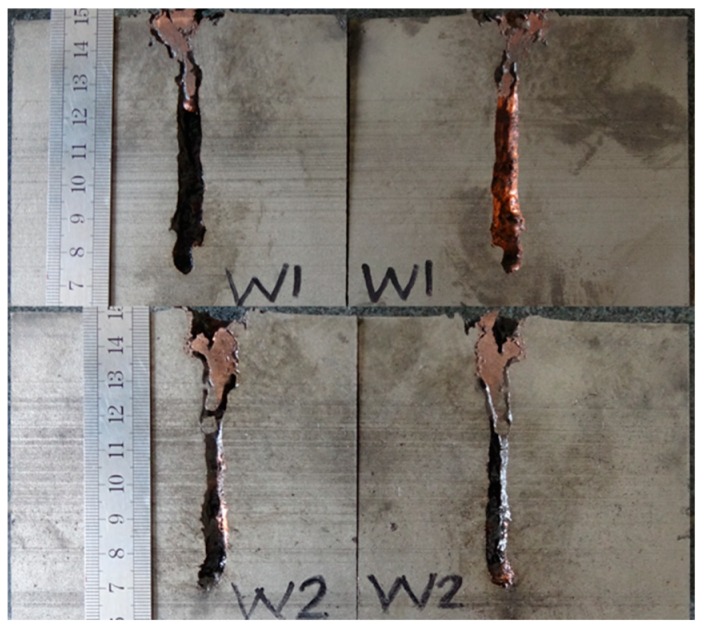
DOP with water as liquid.

**Figure 9 materials-12-01809-f009:**
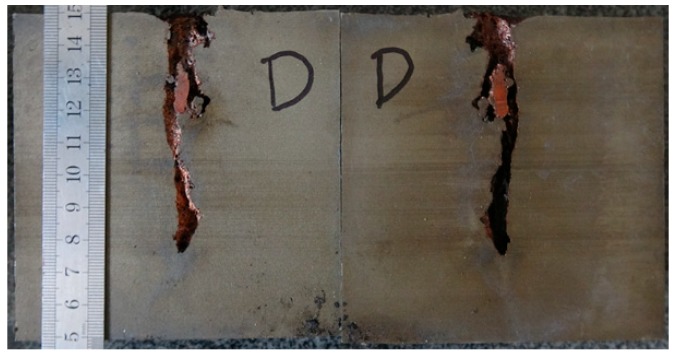
DOP with #0 diesel as liquid.

**Figure 10 materials-12-01809-f010:**
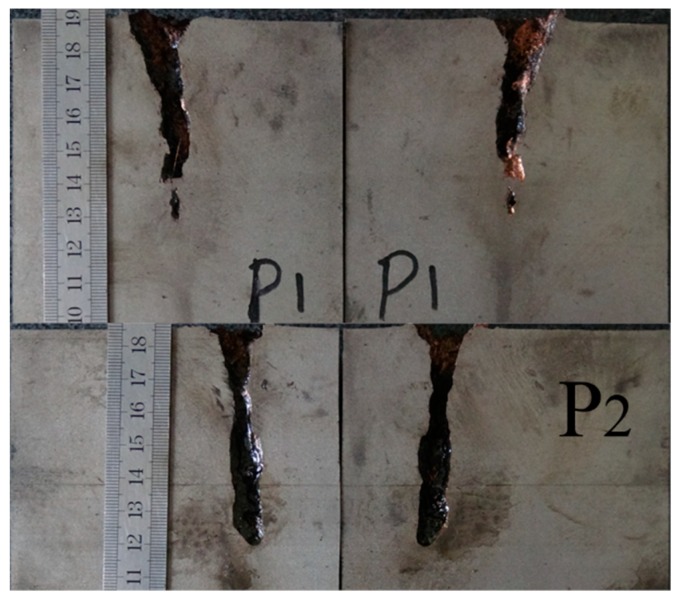
DOP with PEG200 (polyethylene glycol with molecular weights of 200) as liquid.

**Figure 11 materials-12-01809-f011:**
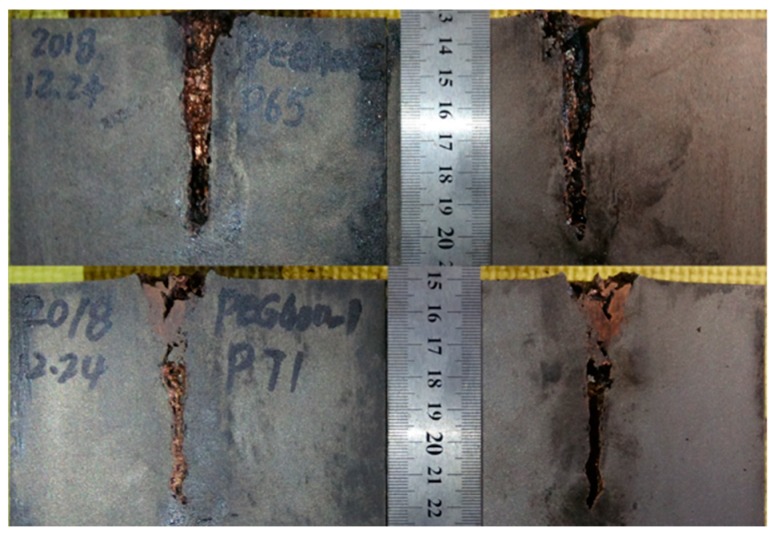
DOP with PEG400 (polyethylene glycol with molecular weights of 400) as liquid.

**Figure 12 materials-12-01809-f012:**
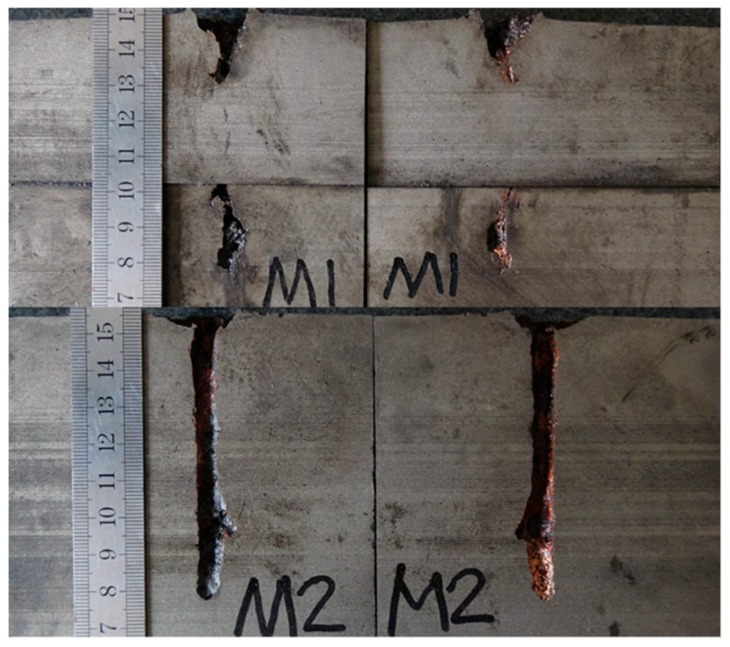
DOP with PPG6000 (polypropylene glycol with molecular weight of 6000) as liquid.

**Figure 13 materials-12-01809-f013:**
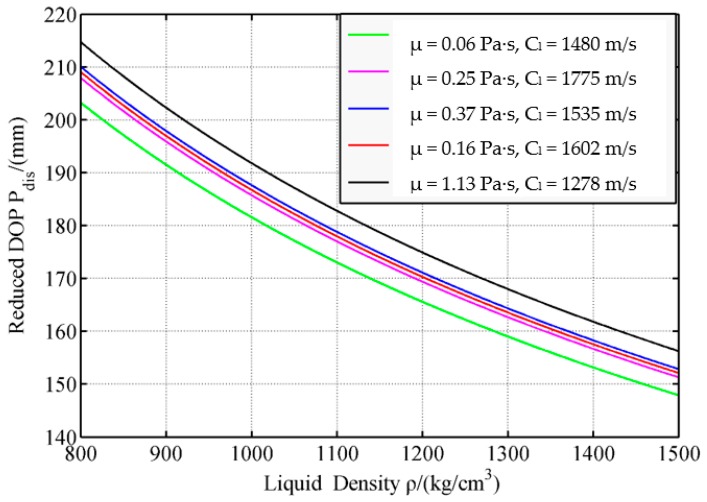
Effects of liquid density on reduced DOP.

**Figure 14 materials-12-01809-f014:**
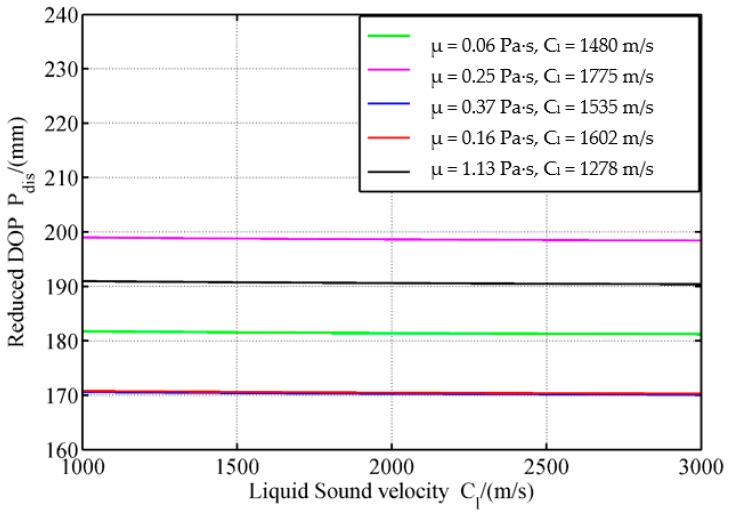
Effects of liquid sound velocity on reduced DOP.

**Figure 15 materials-12-01809-f015:**
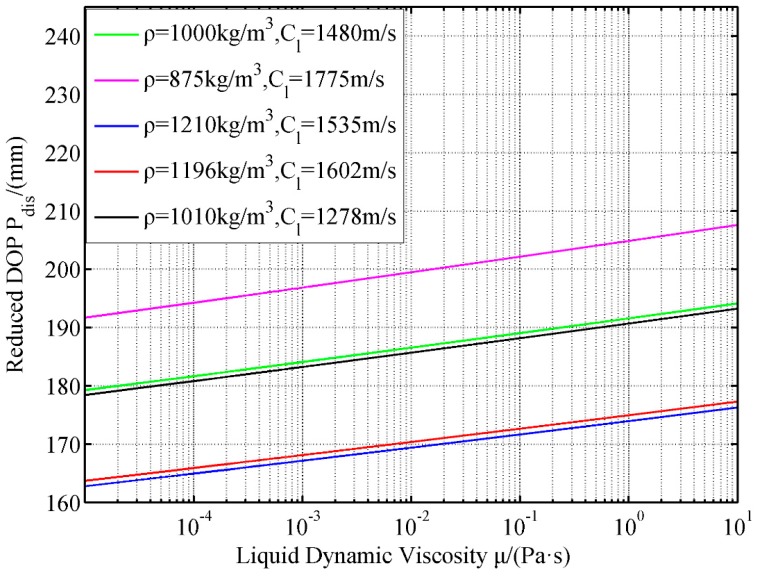
Effects of dynamic viscosity of liquid on reduced DOP.

**Table 1 materials-12-01809-t001:** Main parameters of the liquid, SCJ and target structure.

Material	Parameter	Symbol	Unit	Dimension
Liquid	Density	$\rho_{l}$	$kg/m^{3}$	$M\cdot L^{-3}$
	Sound velocity	$C_{l}$	m/s	$L\cdot T^{-1}$
	Dynamic viscosity	μ	Pa·s	$M\cdot L^{-1}\cdot T^{-1}$
SCJ	Velocity of the jet tip	$v_{j0}$	m/s	$L\cdot T^{-1}$
	Velocity of the jet tail	$v_{jt}$	m/s	$L\cdot T^{-1}$
	Initial length of the jet	$l_{0}$	m	L
	Density	$\rho_{j}$	$kg/m^{3}$	$M\cdot L^{-3}$
	Reduced DOP	$P_{dis}$	m	L
	Sound velocity	$C_{j}$	m/s	$L\cdot T^{-1}$
Compartment structure target	Inner diameter	D	m	L
	Height of the compartment	H	m	L
	Sound velocity	$C_{t}$	m/s	$L\cdot T^{-1}$
	Density	$\rho_{t}$	$kg/m^{3}$	$M\cdot L^{-3}$

**Table 2 materials-12-01809-t002:** Main parameters of the liquid.

Material	μ (Pa·s)	ρ $\left(kg/m^{3}\;\times 10^{3}\right)$	$C_{l}$ (m/s)
Water	0.06	1.0	1480
#0 diesel	0.25	0.875	1775
PEG 200	0.37	1.21	1535
PEG 400	0.16	1.196	1602
PPG 6000	1.13	1.01	1278

**Table 3 materials-12-01809-t003:** Residual penetration in the witness target of SCJ penetrating the LFSC with different liquids.

Liquid	Pres/mm			P_dis_/mm
	First	Second	Average	
Datum	254.0	258.0	256.0	
Water	78.0	81.0	79.5	176.5
Diesel	75.0	-	75.0	181
PEG200	62.0	66.0	64.0	192
PEG400	68.0	71.0	69.5	186.5
PPG6000	72.0	79.0	75.5	180.5

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
