# Peer review of "Effects of Liquid Parameters on Liquid-Filled Compartment Structure Defense Against Metal Jet"

_materials, 2019, doi:10.3390/ma12111809_

Round 1

Reviewer 1 Report

This manuscript reports an investigation on the liquid properties of a liquid-filled compartment in a shaped charge jet (SCJ) through a parameter study supported by experiment. A 1D correlation for penetration depth is calibrated by means of experiments and the effect of viscosity, density and sound speed in different liquids are studied. The conclusions are clearly formulated. This reviewer believes the manuscript reports experiment-based research which can be interesting for the community. However, it suffers from a lack of clarity in many aspects and the way the authors structured their paper has to be revised carefully.  My major concerns are as follows:

-    The presentation of the research is not acceptable and has to be revised. The authors started the dimensional analysis immediately after the introduction. Here, the manuscript lacks an intermediate analytic section to describe the physics (or physical phenomena) first. Also, a schematic of the LFCS could have been shown in this intermediate section. This is essential for clarity of the scientific paper. Otherwise, it would look like a technical report.

-    The equation (5) in the dimensional analysis plays an integral role in the manuscript. However, it is not clear how it is derived from the dimensionless form of equation (3). The assumptions are not fully explained. The proper citation might have been missed. 

-    A general concern goes to the number of experiments for each liquid. I understand the limitations, but the question is if it is enough to draw a general conclusion? 

-   An in-depth discussion on the results (especially figures 12-14) is missing. The authors have only described the figures but not the reason behind such behavior. Also, the abstract is contradictory to the conclusion about the effect of density. 

-    The quality of English has to be improved extensively. There are several long sentences which are hard to follow (e.g. the first lines of the conclusion section). Using some technical terms are incontinent in the manuscript (e.g. dimensionless analysis and dimensional analysis are both used). Also, figure captions have to be more informative. 

Author Response

Point 1: The presentation of the research is not acceptable and has to be revised. The authors started the dimensional analysis immediately after the introduction. Here, the manuscript lacks an intermediate analytic section to describe the physics (or physical phenomena) first. Also, a schematic of the LFCS could have been shown in this intermediate section. This is essential for clarity of the scientific paper. Otherwise, it would look like a technical report.

Response 1: I have add a section (section 2 Physics Phenomena and Principle) to describe the physics phenomena and show the basic principle of the LFCS defence the shaped charge jet. The schematic of the LFCS (Figure 2) have been move to this section at the same time.

Point 2: The equation (5) in the dimensional analysis plays an integral role in the manuscript. However, it is not clear how it is derived from the dimensionless form of equation (3). The assumptions are not fully explained. The proper citation might have been missed.

Response 2: I have added some content to explain how the equation (5) (now it is the equation (6)) is derived from the dimensionless form of the equation (3). And some mistake have been modified at the same time.

Point 3: A general concern goes to the number of experiments for each liquid. I understand the limitations, but the question is if it is enough to draw a general conclusion?

Response 3: There are five group experimental result and the Equation (6) only have three coefficient. I think it can draw a general conclusion. Although it may not have high precision. I have added some discussion the effect of these coefficients in the section 5.

Point 4: An in-depth discussion on the results (especially figures 12-14) is missing. The authors have only described the figures but not the reason behind such behavior. Also, the abstract is contradictory to the conclusion about the effect of density.

Response 4: I have added in-depth discussion on the results after the figures 12-14. The conclusion in the abstract has checked and some sentences have been rewritten.

Point 5: The quality of English has to be improved extensively. There are several long sentences which are hard to follow (e.g. the first lines of the conclusion section). Using some technical terms are incontinent in the manuscript (e.g. dimensionless analysis and dimensional analysis are both used). Also, figure captions have to be more informative.

Response 5: I have modified my paper. The language of this manuscript has been edited by a native English speaker.

Reviewer 2 Report

In the Introduction: 

- rewrite first statement

- Add ref. to Lo Frano et al 2009 at raw 27and similar studies, when referring to local response of a structure undergoing missile impact.  

In section 3:

- rework figure 2 by adding the height of liquid

- information on the instrumentation and measuremets sensors have to be added. In addition uncertainties in teh measurement chain has to be indicated and discussed in order to highlight if it affects the tests measurement 

-Describe, at least synthetically, the test procedure

- in sub-section 3.2 rewite  the unclear statement: "The set experimental standoff is 254 mm (Figure 3). The results of DOP experiments total 254 and 258 mm, whereas the average result of 256 mm is consistently used in this study."

- delete "Figure 6 displays the site layout for the residual penetration experiment." since it is not providing site layout !

-Describe and comment data indicated in table 3

In subsection 3.2, revise the statement "However, comparison results in Tables 2 and 3 reveal that liquid parameters, such as density, sound 176 velocity or dynamic viscosity in the LFCS, show no notable relation with reduced DOP.". I do not agree with it because the force opposing to penetration are depending on the density and viscosity. It is a matter of energy balance, higher the visosity and the density lesser is the penetration (target is stiffer and more prone to penetration, as seemed to be confirmed in statement at raws 187-189) 

In section 4:

- the statement "This result means that at low 187 liquid density in the compartment structure, the structure can disturb a wide jet velocity range, thus 188 increasing the reduction in DOP." seems in contraddiction with the previous at raw 186 and 187

- At raw 190 substitute "protected" with "protection"

- Statement at raw 193 of figure 13 should be revised taking into account the effects of uncertainties.

Author Response

Point 1: In the Introduction:

- rewrite first statement

- Add ref. to Lo Frano et al 2009 at raw 27and similar studies, when referring to local response of a structure undergoing missile impact.

Response 1: I have rewritten the first statement and add ref. to Lo Frano et al 2009 to my paper.

Point 2: In section 3:

- rework figure 2 by adding the height of liquid

Response 2: I have redrawn the figure 2 and add the height of the liquid.

Point 3: Information on the instrumentation and measuremets sensors have to be added. In addition uncertainties in teh measurement chain has to be indicated and discussed in order to highlight if it affects the tests measurement.

Response 3: The instrumentation and measurement sensors are the same as reference 19. So I add the reference in the paper.

Point 4: Describe, at least synthetically, the test procedure

Response 4: I have added the test procedure in the section 4.2 (before is section 3.2).

Point 5: in sub-section 3.2 rewite  the unclear statement: "The set experimental standoff is 254 mm (Figure 3). The results of DOP experiments total 254 and 258 mm, whereas the average result of 256 mm is consistently used in this study."

Response 5: I have rewritten the sentence as “The results of DOP experiments respectively are 254 and 258 mm. so, the average result is 256 mm.”

Point 6: delete "Figure 6 displays the site layout for the residual penetration experiment." since it is not providing site layout !

Response 6:  I have modified that in the paper. Now the setup of the residual DOP experiment is shown as Figure 5.

Point 7: Describe and comment data indicated in table 3

Response 7: I have added more finding and discussion in the reduction of DOP and penetration hole based on Table 3 and Figure 7 to Figure 11.

Point 8: In subsection 3.2, revise the statement "However, comparison results in Tables 2 and 3 reveal that liquid parameters, such as density, sound 176 velocity or dynamic viscosity in the LFCS, show no notable relation with reduced DOP.". I do not agree with it because the force opposing to penetration are depending on the density and viscosity. It is a matter of energy balance, higher the visosity and the density lesser is the penetration (target is stiffer and more prone to penetration, as seemed to be confirmed in statement at raws 187-189) 

Response 8: I agree with the reviewer of this. In the paper, some expression is not clear, so I rewrite them and add some discussion at the same time. Please see the revised manuscript.

Point 9: In section 4:

- the statement "This result means that at low 187 liquid density in the compartment structure, the structure can disturb a wide jet velocity range, thus 188 increasing the reduction in DOP." seems in contraddiction with the previous at raw 186 and 187

Response 9: I have checked that section and rewritten the sentence.

Point 10: At raw 190 substitute "protected" with "protection"

Response 10: I have changed the "protected" with "protection".

Point 11: Statement at raw 193 of figure 13 should be revised taking into account the effects of uncertainties.

Response 11:  I have added some discussion under figure 13. The effect of the diameter of the holes in the structure has been taken into account.

Reviewer 3 Report

The manuscript deals with an interesting topic and presented experimental results that have values to researchers and developers in the corresponding filed. However, the manuscript needs a major revision to be accepted for publication. This reviewer's comments are as follows.

-  In chapter 2, the authors must provided proper references in eqs. (1)-(5). If these are derived by in this research, the derivation must be explained in detail.

- In Fig. 2, provide the vertical view and specify all dimensions of the structure.

- In Table 3, the finding and discussion in the reduction of DOP are qualitative. More in depth discussion of the results must be provided.

- The reliability of the proposed eq. (6) is questionable. The experimental data sets are not enough to determine 3 coefficients which are parts of highly nonlinear behavior. A sensitivity analysis may be done to study the effect of the uncertainties of these coefficients.

- Figs. (12)-(14) do not have any meaningful interpretation. For example, can the density of water be changed between 800-1500 without changing others such as speed of sound and viscosity?

Author Response

Point 1: In chapter 2, the authors must provided proper references in eqs. (1)-(5). If these are derived by in this research, the derivation must be explained in detail.

Response 1: Please provide your response for Point 1. (in red)

Point 2: In Fig. 2, provide the vertical view and specify all dimensions of the structure.

Response 2: I have redrawn the Fig. 2, given the vertical view and all dimensions of the structure.

Point 3: In Table 3, the finding and discussion in the reduction of DOP are qualitative. More in depth discussion of the results must be provided.

Response 3: I have added more finding and discussion in the reduction of DOP and penetration hole based on Table 3 and Figure 7 to Figure 11.

Point 4: The reliability of the proposed eq. (6) is questionable. The experimental data sets are not enough to determine 3 coefficients which are parts of highly nonlinear behavior. A sensitivity analysis may be done to study the effect of the uncertainties of these coefficients.

Response 4: There are five group experimental result and the Equation (6) only have three coefficient. I think it can draw a general conclusion. Although it may not have high precision. I have added some discussion the effect of these coefficients in the section 5.

Point 5: Figs. (12)-(14) do not have any meaningful interpretation. For example, can the density of water be changed between 800-1500 without changing others such as speed of sound and viscosity?

Response 5: We use The Figs. (12)-(14) only want to explain the effect of the parameters of the liquid and find the mainly effect factor, not fixed one kind of liquid. After we find the mainly effect factor and know the better change range of the parameters, maybe we can try to compound a new kind of liquid.

Round 2

Reviewer 1 Report

The authors have addressed/answered almost all of my points and modified the manuscript accordingly. I have no more issue and recommend for publication.

Author Response

(The authors gave the same response as above.)

Reviewer 2 Report

No further comment

Author Response

Thank you for your advice and seriously stringent careful attitude toward work.

Reviewer 3 Report

Response 1: Please provide your response for Point 1. (in red) ==> Is this a request to the reviewer?

Response 4 ==> If you have one coeff. to determine and you have five sets then I will agree. But 3 coeffs to determine and 5 sets of data? If you have a completely known expression, may be your are ok. But first you must prove that the given expression is THE REAL equation, not just a guessed approximation. For the latter, you need to perform a series of independent experiments for each coefficient separately. Then you need to perform level 2 experiments to determine the interactions of the separate behaviors. It is impossible to determine a certain number of coefficients of an expression which is far from the from a compound data set.

Response 5 ==> You could study the effect of parameters. But the range of variation is the problem. Do you actually believe the highly questionable equation will hold for the viscosity range, for example, between 10^-5 and 10?

Author Response

Response to Reviewer 3 Comments

Point 1: Response 1: Please provide your response for Point 1. (in red) ==> Is this a request to the reviewer?

Response 1: I am sorry that I have given more detail about the Eqs. (1) - (5), but forgot to give response here. It is my negligence. Again, express my sincere apologies to you.

Point 2: Response 4 ==> If you have one coeff. to determine and you have five sets then I will agree. But 3 coeffs to determine and 5 sets of data? If you have a completely known expression, may be your are ok. But first you must prove that the given expression is THE REAL equation, not just a guessed approximation. For the latter, you need to perform a series of independent experiments for each coefficient separately. Then you need to perform level 2 experiments to determine the interactions of the separate behaviors. It is impossible to determine a certain number of coefficients of an expression which is far from the from a compound data set.

Response 2: I believe that the three coefficients can be determined by the five groups of experiments. The reasons are as follow:

1) There are three parameters in each kind of liquid, and each parameter of any kind of liquid is different. There are three groups’ parameters and each group’s parameter contains the five variable values (So, in the real measurement, it is impossible for us to finish the DOP experiments when the two parameters need to be fixed with the 3rd one changing).

2) In the present research, we would like to obtain an equation to analyse the effect law of the liquid’s parameters on liquid-filled composite structure to the shaped charge jet through the dimension analysis method.

The Eq.(6) that we have gotten is based on the dimension analysis results rather than the experimental results. We believe that this equation can clarify the influence of the relationship of parameters on the LFCS resist against SCJ. It should be noted that the aim of this paper is to investigate the effect of the liquid’s parameters on LFCS to SCJ (Figures 12- 14 have already proved the discovered effect rule).

3) If the shaped charge or the structure of the LFCS is changed, the coefficients in the Eq.(7) will be varied correspondingly. However, the discovered effect rule of the liquid’s parameters on the liquid-filled composite structure to the shaped charge jet is still suitable.

4) The reasons that the liquid’s parameters make such a change are provided in the section 5.  We believe that the discovered effect rule in this part match with the experimental results.

Point 3: Response 5 ==> You could study the effect of parameters. But the range of variation is the problem. Do you actually believe the highly questionable equation will hold for the viscosity range, for example, between 10^-5 and 10?

Response 3: we think we have already provided the detailed explanations in Point 2, so we can use Eq.(7) to study the effect of parameters. Besides, for the viscosity range (between 10^-5 and 10), the viscosity of liquid in the experiments is within this range.
